# Methods and Experiments for Sensing Variations in Solar Activity and Defining Their Impact on Heart Variability

**DOI:** 10.3390/s21144817

**Published:** 2021-07-14

**Authors:** Michael Hanzelka, Jiří Dan, Zoltán Szabó, Zdeněk Roubal, Přemysl Dohnal, Radim Kadlec

**Affiliations:** 1Department of Theoretical and Experimental Electrical Engineering, Brno University of Technology, 616 00 Brno, Czech Republic; szaboz@feec.vutbr.cz (Z.S.); roubalz@feec.vutbr.cz (Z.R.); dohnalp@feec.vutbr.cz (P.D.); kadlec@feec.vutbr.cz (R.K.); 2Department of Psychology, Faculty of Arts and Letter, Catholic University in Ružomberok, 034 01 Ružomberok, Slovakia; jiri.dan@ku.sk

**Keywords:** ULF, SLF, ELF, BioGraph Infiniti, heart rate variability, sympathetic and parasympathetic

## Abstract

This paper evaluates variations in solar activity and their impact on the human nervous system, including the manner in which human behavior and decision-making reflect such effects in the context of (symmetrical) social interactions. The relevant research showed that solar activity, manifesting itself through the exposure of the Earth to charged particles from the Sun, affects heart variability. The evaluation methods focused on examining the relationships between selected psychophysiological data and solar activity, which generally causes major alterations in the low-level electromagnetic field. The investigation within this paper revealed that low-level EMF changes are among the factors affecting heart rate variability and, thus, also variations at the spectral level of the rate, in the VLF, (f = 0.01–0.04 Hz), LF (f = 0.04–0.15 Hz), and HF (f = 0.15 až 0.40 Hz) bands. The results of the presented experiments can also be interpreted as an indirect explanation of sudden deaths and heart failures.

## 1. Introduction

The low-level measurement of low frequencies (3 Hz–30 kHz) [1] performed to investigate the relationship between changes in the human body and electromagnetic field (EMF) variations can be considered an interdisciplinary branch of science encompassing a variety of research methods and approaches.

The applied research disciplines are very different from the measurement and radar technology in the ULF (Ultra Low Frequency Band: 0.3–3 kHz), SLF (Super Low Frequency Band: (30–300Hz), and ELF (Extremely Low Frequency Band: (3–30 Hz) ranges because they provide statistical methods for processing the impacts and evaluating the signals by which information on human psychophysiology is obtained. The current state of knowledge in the above field is relatively unsatisfactory, [2,3,4], with hasty conclusions having often been made and employed even in the context of legally binding rules or standards (EMF radiation exposure and other limits). An example is the guideline issued by the Council of Europe and implemented by the International Commission on Non-Ionizing Radiation Protection (ICNIR 5P) in 1999, whose purpose was to establish the boundary values of magnetic flux in relation to the rate of magnetic field variation in slowly changing currents (WHO, 2003). Specifically, the guideline introduces the value of 50 mT/s as the maximum magnetic flux alteration acceptable in a human-occupied environment with a variable magnetic field at the frequency of 1 Hz [5]. However, this value is 5–6 orders of magnitude higher than the largest changes measured thus far during the geomagnetic activity disturbances known as magnetic storms. These storms are caused by variations in the intensity of solar activity (eruptions) during which the Earth’s magnetosphere is exposed to charged particles from the Sun, with a portion of the particles penetrating into the Earth’s atmosphere [2]. Geographically, this effect is not limited to the poles but occurs at places below the 70° parallel as well. In 2007, WHO published a monograph entitled *Extremely Low Frequency Fields–Environmental Health Criteria 238* [6] (available at: https://www.who.int/peh-emf/publications/Complet_DEC_2007.pdf, accessed on 8 July 2021), which addresses the potential health effects of human exposure to extremely low frequency electric and magnetic fields. In this context, neither the current version of ICNIRP nor the recommendations by WHO consider the role of naturally and artificially generated low-level fields in functional changes of the human psychophysiological parameters; such changes then arise from minor transformations in the fields. Our article and the underlying experiment therefore focus on HRV fluctuations due to the low-level field induced by changed solar activity.

Since the 1960s, numerous research studies have been conducted worldwide to determine the impact of frequencies such as ULF, SLF, and ELF on the human body, namely, on the nervous and cardiovascular systems [6]. In most cases, the resulting values adhere to ICNIRP guidelines; nevertheless, the research performed by Graham et al. [7] to examine the impact exerted upon human organisms by differently levelled EMFs (low: 6 kV m^−1^ and 10 μT; medium: 9 kV m^−1^ and 20 μT; high: 12 kV m^−1^ and 30 μT) showed that the EMFs induce measurable changes in the human body. For example, significantly lower heart rates were observed among the medium frequency group compared to the other groups, which did not exhibit any heart rate changes. Another study [8], monitoring a set of six physiological parameters, detected longer inter-beat intervals in a similar group of respondents who had been exposed to EMFs with a variation frequency f = 60 Hz.

This paper discusses the procedures and results of experiments measuring a set of psychophysiological parameters in a homogeneous sample of participants, aiming to define the impact of low-level EMF changes on the human body. The EMF variation is caused by solar activity disturbances and has already been scientifically evidenced. The research participants comprised male and female students at the University of Defence (Brno), aged 18 to 25 years, who had qualified for their study by passing strict physical fitness and mental endurance tests. From these students, a homogeneous research sample was chosen [9]. The respondents’ superior physical and mental capacities were expected to naturally produce high resistance of the body to low-level EMF variation (with the magnetic flux densities (B) between 0.01 uT and 1 uT) [9]. Based on the results of the published research available at the moment of the experiment, it appeared reasonable to assume that if measurable changes in physiological data are observed in the above group of participants, the average population, containing a particular percentage of highly sensitive and emotionally responsive individuals, would likely show more pronounced values. The monitored physiological parameters included skin conductance, muscle contraction, and heart rate variability in relation to thoracic vs. abdominal respiration; the relevant data were collected and processed [9] with the aim to confirm or disprove the hypothesis regarding the relationship between the above physiological criteria and geomagnetic field disturbances.

The generation and impact on the human organism of low-level MFs in the context of solar activity has already been investigated within numerous studies, [10,11]; the authors concentrated on the manner in which solar activity variation affected the respondents through changes in the Earth’s magnetic field. The detection of changes and disturbances in the geomagnetic field was performed by means of the Schumann resonances [12,13,14]. The Schumann resonance had been discovered in 1953 by W. O. Schumann of the Technical University of Munich. Until recently, the relevant oscillation was at the average frequency of fsch = 7.83 Hz; however, this value has now been subject to changes in consequence of phenomena such as solar activity variation or greenhouse gases, which alter the Earth-Ionosphere cavity’s resonant properties [4]. During our experiments, solar activity variation was investigated through solar wind monitoring, with the use of data acquired and classified by NASA [11]. It is hypothesised that emotionally unstable, as well as stressed, individuals are generally more sensitive to solar activity changes than those who exhibit greater emotional stability, which leads to heart rate and heart coherence changes in the more sensitive persons. An investigation and analysis of cardiac function data suggested a direct link between increased sudden death and heart failure risks in the sensitive individuals and elevated solar activity. Therefore, the subsequent research in this area concentrated on examining the negative impact exerted upon the human organism by changes in the geocosmic environment, including solar activity variation. Importantly, new diagnostic methods are available to monitor the above phenomena and to evaluate their effects on the human body. The obtained knowledge of the Earth’ s magnetosphere will have a major potential in the area of social behavior prediction, namely, it can aid physical, mental, and stress load management [3].

The research conducted thus far has provided unequivocal evidence that human physiological data and, consequently, social interactions, are affected by solar intensity changes, which can either be monitored directly or through variation in low-level EMF values, those within the 3–30 Hz frequency band in particular.

In addition to the above relationship, there is also evidence [10,11] of mutual interaction between low-level electric or magnetic fields radiated by both the Earth’s geomagnetic system and humans (the electric and magnetic fields emitted by the brain, heart, and other bodily organs).

This paper concentrates upon evaluating the impact on cardiac variability of changes in low-level EMFs within the ELF, SLF, and ELF bands.

## 2. Materials and Methods

The main part of the laboratory-based research rested in collecting extensive psychological data from all participants through several psychological tests, including the ASS-SYM test (from the German Änderungssensitive Symptomliste); this psychometric tool measures the respondents’ sensitivity to the changes in the load-to-rest transition. The instrument comprises six indication areas containing a total of 48 easy-to-understand items (8 items per area); these focus on the symptoms that are sensitive to change in autogenic training and progressive relaxation. The list of symptoms includes general indicators of well-being, relaxation experiences, and discomfort and strain, with no indication of psychopathologies. The six broad areas cover the following factors: physical and mental exhaustion, nervousness and mental tension, psychophysiological dysregulation, performance and behavior-related problems, self-control difficulties, and general symptoms and problems [15]. The results of the above psychological tests were correlated to the outputs of the psychophysiological measurements (from 22 April 2014 to 26 June 2014).

The initial tests were conceived to classify the respondents into groups (Figure 1) based on mental lability. Thus, most importantly, we obtained a set of participants who proved to be indifferent to changes in the intensity of solar activity and the effects of low-level, low-frequency electromagnetic fields; in terms of the classification, these people ranked as mentally stable (group A) [9].

The other groups (indicated in capital letters) and corresponding mental traits were as follows: C represented a high sensitivity to variations in the electromagnetic field, i.e., mental lability; B1 embodied indifference to the monitored changes; and B2 defined increased sensitivity to variations in solar activity. The last of these sets was marked with the descriptor emotionally unstable (or emotionally labile, EL) and became a focus during the evaluation of the research, the main reason for such attention being that an increased level of (LF) had previously been found in patients suffering from posttraumatic stress disorder [16]. In this context, our research results showed that higher (LF) values are produced even by the more moderate psychopathological states, including stress-induced mental lability. To expand somewhat on the actual classification, we can emphasize that the separation into the above groups appeared to be suitable for the purposes of evaluating the measured data; however, the final evaluation involved data acquired generally, from the overall set of participants (the reason being the low total number of respondents).

The laboratory research comprising a homogeneous sample of 49 subjects (male and female, aged 19 to 25 years) began on 22 April 2014 and continued until 26 June 2014. The young age and homogeneity of the research sample ensured stable cognitive competencies such as working memory, selective attention, multitasking, task switching, response monitoring, and error detection. Almost all of these functions showed age-related decline [17]. The total time required to examine the psychophysiological parameters of a subject was 19 min. To carry out the measurements, we used a BioGraph Infiniti unit (Thought Technology, Ltd., Montreal, Canada, https://thoughttechnology.com/ accessed on 8 July 2021) [4,11]. The measurement proper encompassed five phases:(1)(E–Basic)–rest after coming to the laboratory, 5 min;(2)(E–Colour)–mental load involved in naming the color of a text expressing a different color, 2 min;(3)(E–Relaxation)–rest, 5 min;(4)(E–Math)–mental load involved in performing a 2-min mental subtraction of the number 7 from the initial value of 1071; after this mental countdown, the participant was asked to say aloud the result, which was checked for correctness;(5)(E–Rest)–final rest, 5 min.

Two of these measurement stages were stress-based (E-Color and E-Math), while two involved solely relaxation. During the Color stage, a special Stroop test was employed to acquire the psychophysiological data of each subject in response to the mental load imposed on their organism. Generally, this neuropsychological tool demonstrates that people are prone to cognitive interference due to their automatic reactions and reflexes. The procedure is named after John Ridley Stroop (1897–1973), the American psychologist who first described this phenomenon in 1929. The Math phase tested the psychophysiological stress generated by the above-described mental countdown.

The total number of measurements was 210, with the average of 4.29 measurements per participant performed at a time interval of two to five days. Table 1 shows the relationship between the above total count and the number of participants who completed the task. A total of 210 measurements were carried out. The laboratory had been configured to provide a stable testing environment with a constant temperature, noise, humidity, lighting, and concentration of positive and negative ions; also, the geomagnetic field of the laboratory had been homogenized using a pad (see Figure 3c) specially designed to homogenize the surrounding magnetic field and its gradient.

The socioeconomic importance of the research was described in [18] (and presented at a conference in Amsterdam, The Netherlands, from 7 to 8 May 2015; available at https://ibima.org/accepted-paper/measurement-evaluation-effects-low-level-magnetic-fields-socio-economic-behavior-human-body/, accessed on 8 July 2021.

Table 1 shows that the number of completed tests differed between the participants, due to the varying regularity in the respondents’ attendance.

As already mentioned above, the measurements consisted of three relaxation and two stress phases, with an emphasis on intense psychophysiological stress in the participants. The applied stimulation and measurement instruments are presented in Figure 1, and the BioGraph Infiniti experiments are visualized in Figure 2, together with the laboratory equipment. During the stress stages, electromagnetic fields were generated by means of a low-level EMF generator, an amplifier, and Helmholtz coils to interfere with the EEG brain waves of the subjects; the induced low-level EMFs were comparable to the effects of a variation in the Earth’s magnetic field. A generated EMF represented one “layer” of the load that was to be experienced by the participants, enabling us to simulate, via an external device, the effect of solar activity in addition to the real variation in the geomagnetic field. The values of the vector of the magnetic field (H) and the magnetic flux density (B) of the Helmholtz coil were set to the level of the geomagnetic field; they were to approximate the interaction of the geomagnetic field with solar plasma streams. The pulse parameters were as follows: f = 1 kHz; start t = 100 ns; stop t = 168 µs. The magnetic induction (Bd) at a participant’s head corresponded to 1.0 µT; at the laboratory, we utilized a special pad to homogenize the *z* component of the magnetic field and to achieve Bz values of 35–36 µT at the ground level (see Figure 3c; the images in Figure 3a,b) then introduced details of a measuring cycle performed on a respondent).

## 3. Results and Discussion

In the course of the experimental research, a range of psychophysiological data, including skin conductance, muscle contraction, cardiac activity, and EEG brainwaves, were measured with the BioGraph Infiniti software. The basis of the experiment consisted in alternating the mental load and relaxation states, with the mental and health condition of each participant being examined both before and after each session via a psychological test and body temperature and pressure measurements. The obtained data were employed to establish correlations between the human psychophysiological parameters (or the related qualitative psychological measurements) and the intensity of solar activity.

Among the root causes of geomagnetic field variation are charged solar particles (also referred to as solar plasma or solar wind), which stream outward from the Sun; some of them enter the Earth’s magnetic field (magnetosphere). These particles travel, especially in polar areas, along magnetic lines to the upper layers of the atmosphere, where they, in synergy with the ultraviolet radiation from the Sun, become excited and then ionize into neutral atoms. Modern science has developed methods and procedures to measure the changes in solar wind activity, as these are manifested by variation in the low-level geomagnetic spheres [11].

The research covering the interdisciplinary topic of solar activity and its geomagnetic consequences encompasses a vast number of fundamental papers and monographs. In particular, the works by Rollin McCraty [19] and Alexander Tchijevsky [20] are important in many respects: McCraty analyzed the impact exerted by the environment on the physical, mental, emotional, and spiritual coherence of the individual, outlining the relationships between these aspects and the cardiovascular system with its resonant frequency of 0.1 Hz (the ELF band between 0.04 and 0.26 Hz, a prerequisite for cardiac coherence). Tchijevsky, a Soviet biophysicist, found out that 80% of the most significant historical events occurred during the approximately 5 years of increased solar activity, as can be seen in the relevant diagram (1750–1922) [21]. However, no comprehensive investigation has been performed thus far to prove the long-term impact of changes in the Earth’s magnetic field on humans and to exclude the potentially significant effect of other factors.

The presented experimental research monitored solar activity on a daily basis, establishing a set of correlations between the obtained solar-activity values and selected psychophysiological parameters. Solar activity changes reflect the actual variation in the Sun’s magnetic field (i.e., in the activity of solar wind particles), which affects the magnetosphere with a delay of between 2 and 8 days. For this reason, the horizontal axis in the correlation Figure 1, Figure 2, Figure 3, Figure 4, Figure 5 and Figure 6 (see below) represents the number of days since a series of significant solar eruptions; the vertical axis then expresses the relevant correlation index. The actual measurement involved stress stages (E-Color and E-Math), and these embodied the focus of the experiment as regards the variation of HRV parameters; we therefore used a different type of figure in this context. The other stages were relaxation-based, designed to separate the stress phases to ensure sufficient physical and/or mental recovery. In the relevant Figure 1, Figure 2, Figure 3 and Figure 4, the green and blue E-Relaxation columns characterize the conditions following the E-Color and E-Math stress phases, respectively. By extension, a related study [22] proposes that a solar wind could be interpreted as collisionless plasma moving radially from the Sun at speeds oscillating between 300 and 800 km/s. The concentration of the wind varies between 1 and 10 particles per cm^3^, and its temperature ranges between 1 and 30 eV. The mass spectrum of the solar wind includes mainly protons (~0.96%) and helium nuclei (~0.04%), with a small percentage being represented by high-energy electrons.

Turning now to the selected psychophysiological parameters, heart rate variability (HRV) is generally associated with emotional responsiveness [23]. High-frequency (HF) heart activity has been found to decrease in line with acute emotional stress [23,24], and [25], and anxiety. This link is probably related to attention and motor inhibition [26]. The HF component of HRV has been shown to be reduced in people with post-traumatic stress disorder (PTSD), while the low frequency component (LF) is increased [27]. Heart rate variability (HRV) is a physiological phenomenon referring to variation in the beat-to-beat interval. The variation is controlled by the sinoatrial (SA) node located in the wall of the right atrium of the heart, whose activity is influenced by neural mechanisms, namely, the sympathetic and parasympathetic nervous systems (SNS and PSNS, respectively), and humoral factors. The humoral regulation of the circulatory system involves baroreflexes as well as thermoregulatory, hormonal, and stress-related factors. The neural regulatory mechanisms include a decrease in HRV as a result of increased PSNS or SNS activity. High frequency (HF) heart activity (f = 0.15 to 0.40 Hz), above all, is associated with PSNS activity. The above range of heart activity has been linked to respiratory sine arrhythmia (RSA); the normal variation is the heart rate that occurs during breathing and consists in HR increase during inhalation and HR decrease during exhalation. LF variation is presently recognized to reflect the activity of both SNS and PSNS [28].

When correlating the psychophysiological parameters of the human body and changes in solar activity, we have to consider a delay in the impact of such processes onto the Earth. Assuming the average distance between the Earth and the Sun (149,600,000 km), one constituent (protons) of the erupted plasma, the slow solar wind, will reach the Earth’s magnetosphere in 5.77 days (138 h), while the other major component (helium nuclei), the fast solar wind, will complete the same path in only 2.16 days (54 h); the photons will travel at the speed of light *c*, requiring merely 8.31 min.

The flight time of the solar wind particles is defined as follows:(1)$$t_{\min}=\frac{l_{s}}{v_{\min}}$$
(2)$$t_{\max}=\frac{l_{s}}{v_{\max}}$$

The formulas (1) and (2) yield the time interval between the formation of the plasma and the actual impingement of its fast and slow components on the Earth’s magnetosphere (Table 2).

The above-presented simple observations are causally related to magnetospheric changes and their impact on the human organism. In terms of the time characteristics of the individual effects, after a massive eruption on the Sun has been recorded (with the delay of 8.31 min), transformations in the magnetosphere (and, by extension, in the composition of the plasma weight spectrum) can be expected to begin no later than 2.16 days, inducing substantial variations in the electromagnetic fields on the Earth’s surface within 5.77 days from the solar event. According to relevant empirical experiments, however, approximately a 90 % majority of extensive changes appear after eight (−8) days following an eruption [5].

The impact of the above factors on HRV has long been known [20,29]; to date, however, it has not been subject to any investigation in connection with geomagnetic field changes or solar activity variation causing changes in the low-level LF electromagnetic field of the Earth. The referenced papers analyzed the influence exerted upon HRV by solar activity variations, using the Pearson correlation coefficient to create a set of dependencies reflective of the relationships between the given variables. Figure 4 presents the correlation dependency of the psychophysiological parameter B (basically HRV; see below): BVP peak freq. mean (Hz). The HRV corresponds to fv = 0.04–0.15 Hz; fv > 0.15 Hz is indicative of parasympathetic activity [30]. The maximum correlation occurred with a delay of 8 days after (the minus sign at relevant spots in the text means 1, 2, or 3 days before) a solar wind peak ejected into the magnetosphere a stream of particles that affected the parameters of the respondents. Here, parasympathetic activity (the HF component) prevailed. A noteworthy aspect of this relationship consists in the negative correlation represented by the red curve, which expresses the mental and physical strain imposed on the participants by the Stroop color test (see above). The workload increased the respondents’ heart rate as a result of parasympathetic suppression in connection with the HF component decrease. Psychophysiologically, the Stroop color-naming test shifted the HRV into a certain borderline zone characterized by HRV changes; namely, the HRV decreased together with the HF component. A logical consequence of the HF component reduction was an increase in the LF constituent; however, the hypothesis concerning correlation dependence of heart rate variability on solar activity was not confirmed (Figure 3). If solar disturbances do affect HRV during the relaxation phase, the above lack of correlation can indicate that the heightened mental and physical workload overrode the hypothesised effect (Figure 4, Figure 5 and Figure 6). Another significant aspect is expressed by the yellow curve in Figure 1, which illustrates the marginal influence of the parasympathetic nervous system during the second (E-Math, or “countdown”) test; this can be explained by that the color task was more immersive and activated several different functions at a time (vision and speech), thereby subjecting the participants to relatively high stress. The E-Math test, by contrast, consisted in performing mental countdown and only involved cognitive brain functions, not increasing parasympathetic activity; therefore, no heart rate acceleration was observed. These results indicate that low-level, low-frequency EMF changes induced by solar activity variation increased the HF component of heart rate variability and simultaneously decreased the LF component, to an extent depending on the overall mental and physical condition of the subject.

Figure 5 illustrates the correlation dependency of the psychophysiological parameter B (BVP HF % power mean) on the intensity of solar activity, showing that the correlation reaches its maximum at approximately t = −8 to −9. In the given context, we have to consider the first major peak relating to the electromagnetic impact of the solar wind; at the time of the experiment, this peak occurred on or slightly after the sixth (−6) and, in terms of the HRV impact, culminated between days −8 and −9. With increasing influence of the low-level, low-frequency EMF, HF stimuli begin to prevail, thus affecting the parasympathetic nervous system (PSNS).

Figure 6 shows the correlation dependency of the psychophysiological parameter B (BVP LF % power mean) on solar activity changes. In the figure, no correlation maximum can be observed around the value of − at t = −8, which indicates that the correlative relationship has negligible significance; in other words, LF stimuli cannot be said to prevail with increasing influence of the low-level, low-frequency EMF, and the parasympathetic nervous system is not affected.

Figure 7 shows the correlation dependency of the psychophysiological parameter B (BVP VLF % power mean) on solar activity variation. A correlation maximum can be seen to occur at t = −8 (with a culmination between days −6 and −8), conveying a strong relationship between the two variables. The VLF band is another significant indicator of sympathetic activity. A relevant study [31] revealed a VLF band increase in obstructive sleep apnea (OSA) patients, with the elevation being concurrent with hypoxemia (respiration arrest) episodes; thus, the VLF band was found to be associated with low levels of oxygen in blood. According to the research, the VLF band of heart rate variability in OSA patients is connected to internal oscillation in the vasomotor part of the baroreflex loop and can therefore be considered a direct result of sympathetic activity. The findings by Hadase et al. are in line with the results of our E-Color test, which show the strongest correlation between the observed variables: the physical load imposed by the reading out of colors increased lung ventilation. During the other experimental phases (E-Basic, E-Math, E-Relaxation), no significant correlation was observed between the VLF band of the HVR and solar activity variation.

Another investigated parameter was the intrinsic frequency of the SA node (IFS), which was found to exhibit very weak correlation to solar activity disturbances; nevertheless, a correlation maximum can be observed at t = −8. The intrinsic cardiac frequency (or the intrinsic frequency of the sinoatrial node, IFS) is determined by the activity of pacemaker cells in the sinoatrial node, independently of the autonomic nervous system, and declines with age. IFS values can be calculated using the following equation: IFS = 118.1 − (0.57 × age). The factors affecting IFS can be of physical (mechanical influence of the respiration process, temperature) or chemical nature; also, they include blood supply of the sinoatrial node [32]. Clearly, IFS shows dependence on solar activity variation (see Figure 8), which holds true for the entire research sample and all the individual (stress and rest) measurement stages. Our results are congruent with the defining characteristics of IFS, proving once more that IFS variation, albeit affected by solar intensity changes, does not exhibit any dependence on the parasympathetic nervous system (PSNS). Therefore, any increase in IFS due to solar variation was likely the result of elevated heart rate incited by increased sympathetic activity.

The low correlation indices lead to insufficient confirmation of the research hypotheses. For this reason, no association analysis was conducted (e.g., via χ2) to confirm or disprove the H0 hypothesis [9] suggesting the existence of correlational dependency of heart rate variability on solar activity disturbances. Nevertheless, it needs to be considered that the examined set of correlational dependencies constitutes a part of a profoundly complex and dynamic system of mutual relations involving solar activity, the geomagnetic field, and the human organism, complete with their synergies and interactions (or a lack thereof); these factors can then combine to impact the functioning of economic units, which can be reflected in, for instance, capital market fluctuations manifested by variation of stock indices [33] DJIA (Dow Jones Industrial Average) and [34] S&P 500).

During the experimental research, we employed the Biograph Infiniti measurement platform to obtain a sufficient amount of quantifiable psychophysiological data yielded from the relaxation and mental stress states in the respondents. In this context, psychological tests facilitated determining the mental condition of the participants, and the relevant results were expressed qualitatively. The adopted methods and methodologies allowed us to form a unique, broad set of correlations between human psychophysiological parameters and changes in the intensity of solar activity on the one hand and the corresponding secondary variations in the external magnetic (geomagnetic) field on the other.

The evaluation presented herein relates to participants whose test results (ASS-SYM, MBTI, Lüscher–Color–Diagnostik, and mirror drawing) exhibited increased mental lability.

## 4. Conclusions

The methods, methodologies, and results allow and are yielded from experiments conceived to characterize the impact of changes in the intensity of solar activity on a human being and their social interactions. In this context, the relevant primary research showed that geomagnetic field variations due to solar flares affect the neuro- and psychophysiological factors enabling an individual to successfully integrate themselves and perform within society. Significantly, the revealed relationships between the analyzed natural effects and human mental and physical states have not been proved elsewhere to date. In electromagnetic terms, the experiments exposed the role of the low-level, low-frequency portions of the external field, namely, the magnetic component in the frequency spectrum region of f = 3–30 Hz. The adopted methodologies also allowed us to evaluate the parameters of the interactions between the quantities and processes involved, producing a dataset broadly usable in further, expanded investigations of the outlined problems.

In further experimental research, we will focus on sensing human EEG activity and its correlation with low-level electromagnetic fields. For this we will use sensors for remote data transmission, for example according to [5].

The knowledge and conclusions acquired during the research are essential for the overall application of predictive control in social and economic processes, typically within the health sector, transportation, industry and finance markets, supply of energy and goods, and safety or security.

The planned future activities will involve, among other tasks, further evaluation of the above-outlined relationships in all of the participants, not excluding those with a stable (or less variable) mental state. Classifying the respondents into groups, in addition to the designed procedures, will allow us to set up the criteria to define the psychophysiological parameters in the healthy, emotionally stable individuals.

## Figures and Tables

**Figure 1 sensors-21-04817-f001:**
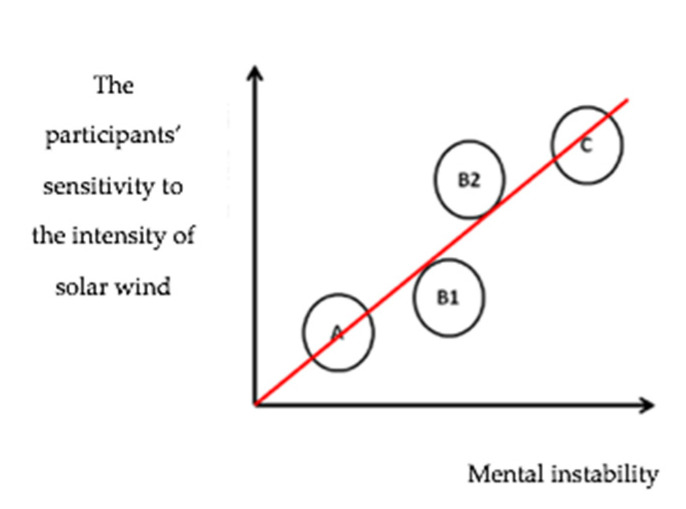
The relationship between the respondents’ mental sensitivity and the intensity of solar activity [9].

**Figure 2 sensors-21-04817-f002:**
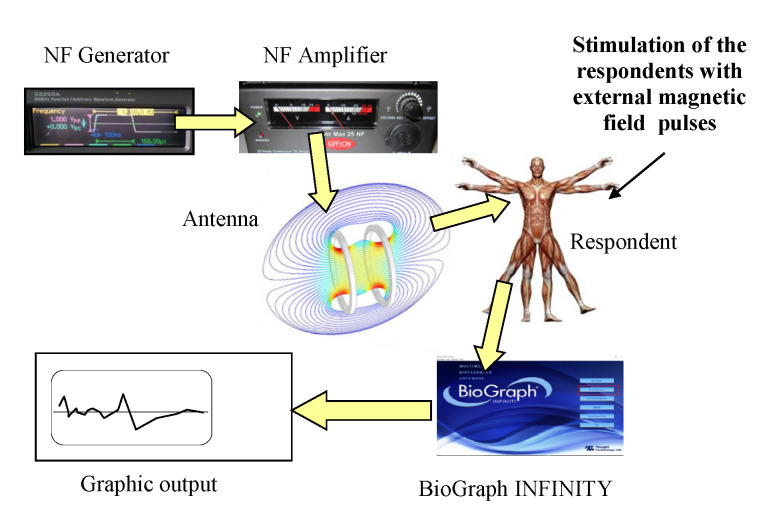
The measurement and simulation equipment.

**Figure 3 sensors-21-04817-f003:**
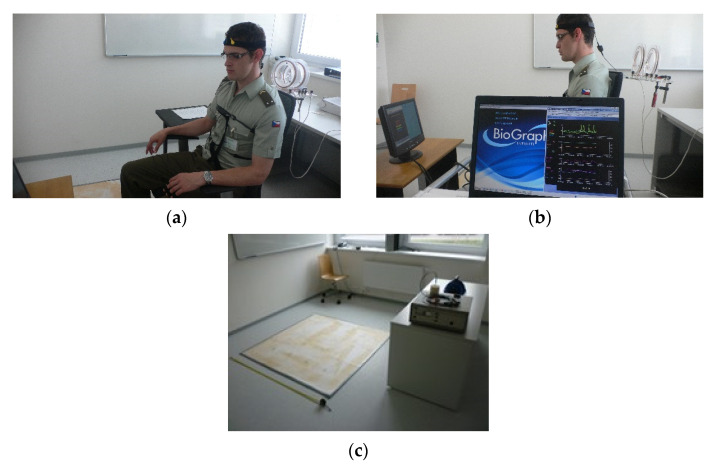
(**a**–**c**) Images of the laboratory measurements, (**a**,**b**); the room with a geomagnetically stable magnetic field component, (**c**).

**Figure 4 sensors-21-04817-f004:**
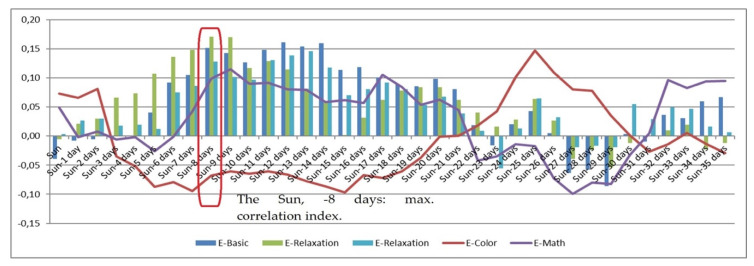
The correlation dependency of the psychophysiological parameter B (BVP peak freq. mean (Hz) (HRV)) on the intensity of solar activity.

**Figure 5 sensors-21-04817-f005:**
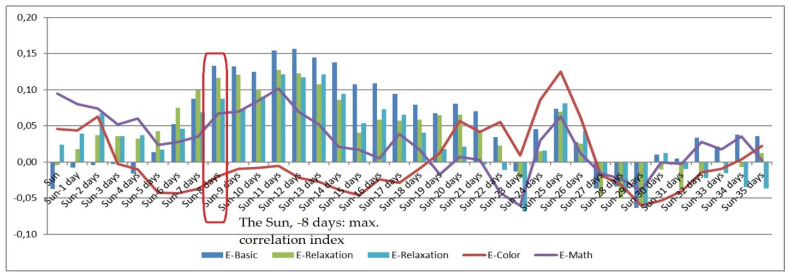
The correlation dependency of the psychophysiological parameter B (BVP HF % power mean) on the intensity of solar activity.

**Figure 6 sensors-21-04817-f006:**
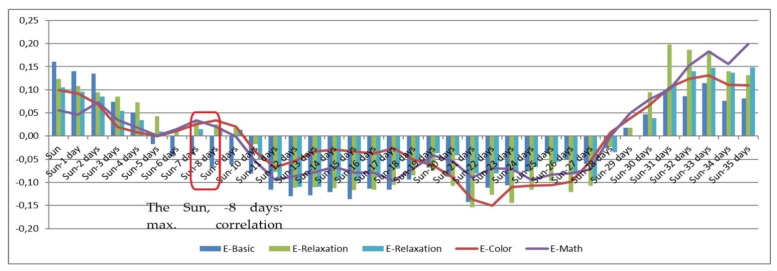
The correlation dependency of the psychophysiological parameter B (BVP LF % power mean) on the intensity of solar activity.

**Figure 7 sensors-21-04817-f007:**
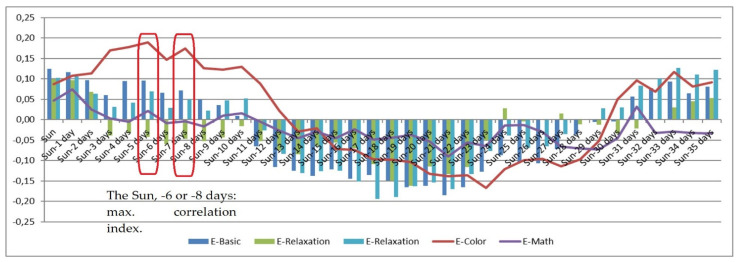
The correlation dependency of the psychophysiological parameter B (BVP VLF % power mean) on the intensity of solar activity.

**Figure 8 sensors-21-04817-f008:**
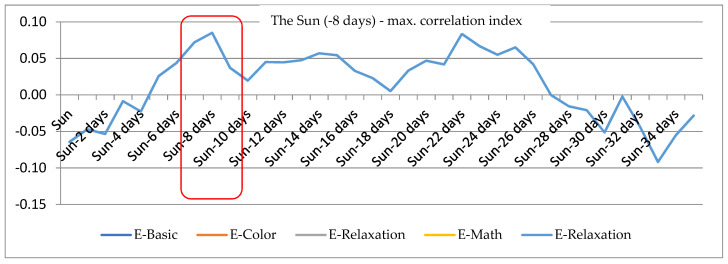
The correlation dependency of the psychophysiological parameter IFS on the intensity of solar activity.

**Table 1 sensors-21-04817-t001:** The relationship between the number of completed measurements and the count of respondents.

Number of Respondents	Number of Measurements
4	1
1	2
4	3
12	4
24	5
4	6
49	210

**Table 2 sensors-21-04817-t002:** The times, speeds, and distances relating to solar wind generation and impingement.

Quantity	Value
Average distance Earth–Sun (km)	149,600,000
Min solar wind speed (km/s)	300
Max solar wind speed (km/s)	800
*t*_min_ = *l*_s_/*v*_min_	5.77 days
*t*_max_ = *l*_s_/*v*_max_	2.16 days
*t*_f_ = *l*_s_/*c*	8.31 mins

## Data Availability

Some research outputs are publicly available here: https://www.vutbr.cz/en/students/final-thesis/detail/104368 accessed on 8 July 2021.

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
