# Peer review of "Methods and Experiments for Sensing Variations in Solar Activity and Defining Their Impact on Heart Variability"

_sensors, 2021, doi:10.3390/s21144817_

Round 1
Reviewer 1 Report
The present study results are highly important for many researchers, from physics, engineering to psychology. Do EMF changes induced by sun influence heart-rate variability and in turn cognitive performance. The authors answer decisively with 'yes'.
Unfortunately, the paper, as intersting as it is, is rather chaotically structured. This had to improve considerably.
1. Parts of the text (paragraphs) which would be labeled Theory, Methods, Results, Discussion are completely mixed. For example, in the chapter "Procedures and methodology" on page 6 you find Results (lines 227-230), Theory (lines 236-241 plus first paragraph on page 7; or it could be part of Discussion). Somewhere from lines 223 on actually this chapter should be termed Results. But the heading of Results appears only on page 10 after one has had 4 pages of results already. The Conclusion section on page 13 does not deserve its name. In what is termed the Discussion section on page 12, sudddenly new results (Graph 9) suddenly appear. This is very confusing.
2. The Results (probably starting somewhere at lines 227) are also confusing as terms are explained in an unsatisfactory way. Lines 229 explain vertical axis as "relevant correlation index". What is meant is that there is the correlation between solar activity (explicate the exact measure; probably it is somewhere but hidden) and the five measurement phases. Next: why are two phases (E-Color, E-Math) depicted as lines and the other three as boxes? The values are on the y-axis and are the correlation coefficient r (I had to deduce this), label it correctly.
3. Results (mainly graphs 1 to 5): sometimes one sees that day +8 is linked to a first peak and sometimes not. This seems arbitrary or stochastically random. I would suggest to argues more clearly (mathematically?) why day +8 seems so significant to the authors. The results are again presented in random order, i.e. not systematically, on page 7, Fig. 3 is first described, and then Fig. 1. Why this non-serial order? Could one have sub-headings in this results section (now wrongly labeled "Procedures and methodology") for the different parameters?
4. I would delete the section labelled as "3. Results". It is not introduced in the introduction and is a different type of study (a retrospective data comparsion study) not much related to the first prospective study with experiments on humans. While the prospective study can be defended by its empirical methodology, the second will face lot of criticism as it is more dependent on post-doc interpretations. While this is ok in itself, it does not fit in here and would deserve a second paper with its own introduction and theory, which is more or less absent here or sqeezed in.
Author Response
Dear Referes,
Thank you for sending the manuscript to the referees. The reviewers' comments proved very valuable in that they allowed us to organize the individual text passages and explain the less comprehensible parts of the text in a more acceptable way.
We have incorporated your comments into the revised text according to your recommendations.
In the revised version, the answers to the referee's comments are highlighted in color.
Thank you for considering the modified manuscript and the request.
Kind regards
Michael Hanzelka

Reviewer 2 Report
General comments:
The authors are aiming to define how low-level EMF changes impact the human body, as well as to connect solar activity with stock market changes. The manuscript discusses procedures of experimental measuring and obtained results. The evaluation methods focused on examining the relationships between selected psychophysiological data and solar activity. The research revealed that low-level EMF changes induced by solar activity variation increase the high-frequency component of heart activity and decrease the low-frequency component, while no frequency correlation was observed between the very-low-frequency band and solar activity variation.
Although the results give important findings of the influence of the low-level EMF on the human body, the paper needs to be improved to emphasize the most important findings. The description of the research methodology needs to be clarified. Also, there are a lot of the same sentences in the manuscript as in the manuscript of the same authors with DOI: 10.1515/msr-2017-0005. Those sentences should be reformulated. The manuscript can be considered for publication after revisions on the issues listed in Specific comments.
Specific comments:
The manuscript should be prepared according to the Instructions. Back matter should be added. Table and figure captions should be formatted according to instructions (Figure 1, Scheme I, Figure 2, Scheme II, Table 1, etc.). References should be formatted according to the journal’s Reference Formatting Guide.
1.Introduction
In the first and second chapter, there is a lot of the same text, written using the same or similar sentences, as in the manuscript with DOI: 10.1515/msr-2017-0005 (such as 31-32, 34-35, 38-39, 86-87, 117-120, Tab. 1, Fig. 1c, etc.). However, that manuscript is not cited at all.
Lines 38-42
This manuscript:
“Specifically, the guideline introduces the value of 50 mT/s as the maximum magnetic flux alteration acceptable in a human-occupied environment with a variable magnetic field at the frequency of 1 Hz. However, this value is 5-6 orders of magnitude higher than the largest changes measured thus far during the geomagnetic activity disturbances known as magnetic storms.”
The manuscript with DOI: 10.1515/msr-2017-0005:
“More concretely, this guideline introduces the value of 50 mT/s as the maximum magnetic flux change acceptable in an environment having a variable magnetic field at the frequency of 1 Hz and characterized by the permanent presence of humans. This value is many million times higher than the largest changes hitherto measured during processes referred to as magnetic storms, in which the Earth was exposed to charged particles from the Sun.”
Please explain.
Please, add a reference for the second statement (“this value is 5-6 orders of magnitude higher than the largest changes measured thus far during the geomagnetic activity disturbances”).
Line 77
Reference is listed twice.
- Methods for the measurement and monitoring of selected physiological parameters
The title of Chapter 2 is inconsistent with the content of the chapter. If I understood correctly, the chapter describes ways to measure psychological and psychophysiological parameters but also talks about results described in lines 254-364. Also, some other parts of the text in chapter 2 text have no place in the chapter describing the methodology, such as 203-222, 236-253, etc. In the chapter named methodology just procedures and methodology should be written, nothing else.
Overall, chapter 2 needs to be completely reorganized and reformulated because it is very difficult to understand what was measured and at what point, and how the authors assessed the impact of solar activity on psychophysiological parameters. I suggest changing the title of chapter two to “Materials and Methods”.
Line 144
Fig.2c does not exist. Did you mean to Fig 1?
Lines 146-149
Please add few basic conclusions of the research presented at the conference.
Line 151, Tab. 1
To make the table more readable, please write the number 210 instead of the word SUM and separate or emphasize the last row.
Lines 152-155
Please explain how many respondents were taken into consideration? Is it 49, or 24, or 24+12?
In lines 419-420 you claim that the presented evaluation relates to participants who have increased mental lability, so it remains unclear based on how many participants out of 49 an analysis was performed whose results are presented in the paper.
Lines 254-256 and 259-261
Sentences are not understandable. Please reformulate.
Lines 254-364
Graphs 1-5, and the text associated with them should be in the Chapter Results and discussion.
What is the difference between green and blue E-Relaxation on graphs 1-4?
Please define and describe in one place in the text all categories of heart rate oscillations (ULF, VLF, LF, HF).
Graph 5. There are 5 parameters in the legend and one in the graph.
3.Results
Line 378
The 20th solar cycle ended at the end of the year 2019, so it is not current.
Line 392-396
In Graph 8, the arrow points to the year 1986, while the text on lines 392-369 talks about 2008.
Can you connect some other historical deep falls on stock markets with solar minimums?
4.Discussion
Lines 440-446
The planned future activities should be written into Conclusions.
Author Response

(The authors gave the same response as above.)

Round 2
Reviewer 1 Report
The manuscript has improved in that it is more concisely structured and to the point.
A quuestion: In the graphs, why are the days post solar activity labaled in increasing days with a minus sign and not positive. It is about post-solar activity, should it not be days 1, 2, 3, etc. and not days -1, -2, -3, etc.?
Author Response
Comments and Suggestions for Authors
The manuscript has improved in that it is more concisely structured and to the point.
A quuestion: In the graphs, why are the days post solar activity labaled in increasing days with a minus sign and not positive. It is about post-solar activity, should it not be days 1, 2, 3, etc. and not days -1, -2, -3, etc.?
MH: explained on line 324

Reviewer 2 Report
Line 355
There is no yellow curve in fig. 1.
Graphs 1-4
The authors did not answer the question: "What is the difference between green and blue E-Relaxation on graphs 1-4?" I think that should be specified in the text.
Graph 5
The authors claim to have explained why there are 5 parameters in the chart legend and only one on the chart itself (in lines 418-419), however, it is still not clear to me why this is so. In my opinion, it is necessary to revise the chart so that the reader knows exactly what the blue line shows.
The Discussion section is very sparse and represents a recapitulation of what has been done. In my opinion, the section "Results" and the section "Discussion" should be replaced by one chapter "Results and discussion".
Line 34
Reference 37 is added after reference 3 (same for reference 38 in line 42). Should they be sorted in order of occurrence?
Lines 258-261
The text is underlined.
References
The citation style is not uniform.
Graphs and figures
In my opinion, figures and graphs should not be marked separately but only as figures.
Author Response
Comments and Suggestions for Authors
Line 355
There is no yellow curve in fig. 1.
MH: It is not clear to me what the yellow curve is, graph No. 1, as far as the green one is concerned, it is described on line 263
Graphs 1-4
The authors did not answer the question: "What is the difference between green and blue E-Relaxation on graphs 1-4?" I think that should be specified in the text.
MH: Explained on line 263
Graph 5
The authors claim to have explained why there are 5 parameters in the chart legend and only one on the chart itself (in lines 418-419), however, it is still not clear to me why this is so. In my opinion, it is necessary to revise the chart so that the reader knows exactly what the blue line shows.
MH: Graph 5 has been revised
The Discussion section is very sparse and represents a recapitulation of what has been done. In my opinion, the section "Results" and the section "Discussion" should be replaced by one chapter "Results and discussion".
MH: Explained on line 224
Line 34
Reference 37 is added after reference 3 (same for reference 38 in line 42). Should they be sorted in order of occurrence?
MH: It has been renumbered
Lines 258-261
The text is underlined.
MH: removed
References
The citation style is not uniform.
MH: removed
Graphs and figures
In my opinion, figures and graphs should not be marked separately but only as figures.
Submission Date
07 June 2021
Date of this review
29 Jun 2021 13:11:30
